# Manganese Ferrite Nanoparticles (MnFe_2_O_4_): Size Dependence for Hyperthermia and Negative/Positive Contrast Enhancement in MRI

**DOI:** 10.3390/nano10112297

**Published:** 2020-11-20

**Authors:** Khairul Islam, Manjurul Haque, Arup Kumar, Amitra Hoq, Fahmeed Hyder, Sheikh Manjura Hoque

**Affiliations:** 1Materials Science Division, Atomic Energy Centre, Dhaka 1000, Bangladesh; khairul.aeceiu@gmail.com (K.I.); arup.mb@gmail.com (A.K.); anhoq120@gmail.com (A.H.); 2Department of Electrical and Electronic Engineering, Islamic University, Kushtia 7003, Bangladesh; manju_iu@yahoo.com; 3Department of Radiology & Biomedical Imaging, Magnetic Resonance Research Center (MRRC), Yale University, New Haven, CT 06519, USA; fahmeed.hyder@yale.edu

**Keywords:** manganese ferrite, X-ray diffraction, nanomaterials, specific loss power, magnetic resonance angiography

## Abstract

We synthesized manganese ferrite (MnFe_2_O_4_) nanoparticles of different sizes by varying pH during chemical co-precipitation procedure and modified their surfaces with polysaccharide chitosan (CS) to investigate characteristics of hyperthermia and magnetic resonance imaging (MRI). Structural features were analyzed by X-ray diffraction (XRD), high-resolution transmission electron microscopy (TEM), selected area diffraction (SAED) patterns, and Mössbauer spectroscopy to confirm the formation of superparamagnetic MnFe_2_O_4_ nanoparticles with a size range of 5–15 nm for pH of 9–12. The hydrodynamic sizes of nanoparticles were less than 250 nm with a polydispersity index of 0.3, whereas the zeta potentials were higher than 30 mV to ensure electrostatic repulsion for stable colloidal suspension. MRI properties at 7T demonstrated that transverse relaxation (T_2_) doubled as the size of CS-coated MnFe_2_O_4_ nanoparticles tripled in vitro. However, longitudinal relaxation (T_1_) was strongest for the smallest CS-coated MnFe_2_O_4_ nanoparticles, as revealed by in vivo positive contrast MRI angiography. Cytotoxicity assay on HeLa cells showed CS-coated MnFe_2_O_4_ nanoparticles is viable regardless of ambient pH, whereas hyperthermia studies revealed that both the maximum temperature and specific loss power obtained by alternating magnetic field exposure depended on nanoparticle size and concentration. Overall, these results reveal the exciting potential of CS-coated MnFe_2_O_4_ nanoparticles in MRI and hyperthermia studies for biomedical research.

## 1. Introduction

The applications of nanomaterials in the biomedical field allows solving many issues such as targeted drug delivery [1,2], contrast-enhancing dye in magnetic resonance imaging (MRI) [3,4,5,6,7,8], mediators for hyperthermia applications [9,10,11,12,13,14], cell labeling and tracking [15], angiography with MRI [16,17,18], cellular transfection using magnetic fields [19], cerebral blood volume (CBV) experiments of functional MRI (fMRI) [20], drug distribution in the brain [21], and antimicrobial activity agent [22]. Surface functionalized/modified spinel ferrite nanoparticles such as MnFe_2_O_4_, MgFe_2_O_4_, CoFe_2_O_4_, ZnFe_2_O_4_, Fe_3_O_4_ are excellent mediators for cancer thermotherapy and MRI contrast agents [23,24,25,26,27]. These nanoparticles are biocompatible, biodegradable, possess high transition temperatures, and have excellent chemical stability. Moreover, nanomagnetism of ferrite nanoparticles provides the opportunity for several biomedical applications because these possess higher magnetic susceptibility than normal superparamagnetic materials and negligible coercivity (i.e., field needed to demagnetize) and retentivity (i.e., residual magnetism after field removal).

Properties of ferrites depend on their composition and microstructure, which in turn depend on their synthesis processes. There are various chemical and physical methods [28,29,30,31,32,33,34,35,36,37,38,39,40,41] to synthesize ferrite nanoparticles, such as chemical co-precipitation, sol-gel auto combustion (i.e., combustion of solution of metal salts and organic fuel forms a gel), reverse micelle, microwave hydrothermal, sonochemical, forced hydrolysis, one-step, high energy ball milling, solvothermal, and microemulsion method. Chemical co-precipitation has several advantages over others, such as (i) uniform and homogeneous nanoparticles of semi-spherical sizes, (ii) control of particle size by varying the reaction parameters such as reaction temperature and pH of the solution, (iii) composition flexibility, (iv) facile, and (v) large scale preparation technique. The co-precipitation method involves the simultaneous occurrence of nucleation at several locations and inhibits the growth mechanism. Since this process requires less heat, only about 80 °C for ferritization reaction, particle sizes are relatively smaller than any other synthesis method.

MnFe_2_O_4_ nanoparticles are of great interest for their remarkable inherent biocompatibility because of the presence of Mn^2+^ ions, tunable magnetic properties, higher transition temperature, and excellent chemical stability for room temperature applications. There are recent reports on MnFe_2_O_4_ nanoparticles as one novel agent for magnetic hyperthermia and MRI contrast [42,43,44,45]. Manganese based ferrites have added advantages than other cations because manganese can be consumed in the amount of 0.67–4.99 mg with a mean value of 2.21 mg per day [46].

Magnetic particle hyperthermia is a very efficient technique for localized destruction of cancer cells, targeted drug delivery, and synergistic use of hyperthermia with chemotherapy and radiotherapy [47]. Specific loss power (SLP), i.e., the ease with which nanoparticles heat the surrounding media, depends on Néel and Brownian relaxation mechanisms and the hysteresis loss [48]. Recent studies suggest that the contributions of hysteresis losses related to the areas of the hysteresis loops are vital for SLP. The SLP also depends strongly on the magnetization and anisotropy of the nanomaterials. In the nanoparticulate systems, where the conventional size law breaks down, magnetization and Curie temperatures are no more intrinsic properties of materials; rather, they depend strongly on the nanoparticle size and surface functionalization/modification. MnFe_2_O_4_ possessing divalent Mn^2+^ cations have five unpaired electrons in the d-orbitals resulting in a magnetic moment of 5. Since Fe^3+^ ions are antiparallel in the tetrahedral and octahedral sites, the uncompensated magnetic moments of Mn^2+^ provide reasonable magnetic moments for higher SLP, which can further be tuned by modifying nanoparticle size. Further, nanoparticle size change also incurs shape anisotropy that has a strong influence on SLP [11,48,49]. In the monodomain range, an increase of nanoparticle size would reduce anisotropy since anisotropy is inversely proportional to the nanoparticle size. Limited value of anisotropy enhances SLP, beyond which it is detrimental because higher anisotropy inhibits Néel and Brownian relaxation mechanisms.

Ferrites are ideal MRI contrast agents for higher transverse relaxation time (T_2_), and MnFe_2_O_4_ is one among them [45]. However, recently, applications of MnFe_2_O_4_ are explored as the contrast agent for magnetic resonance angiography (MRA). The MRA sequences, which use low repetition and echo time, create images that are weighted by longitudinal relaxation time (T_1_). Mn^2+^ cation has five unpaired d electrons that give rise to the magnetic moment, which increases T_1_ relaxation, and therefore, possesses higher T_1_ relaxation [50]. Mn^2+^ based compounds such as MnO, Mn_3_O_4_, Mn_3_O_4_@SiO_2_, hollow MnO are some of the recent developments of Mn-based T_1_ agents. Though MnFe_2_O_4_ is a T_2_ contrast agent, Zhang et al. [51] demonstrated that MnFe_2_O_4_ enhances the T_1_ relaxation when the nanoparticle sizes are exceedingly small and monodisperse.

In this work, we synthesized MnFe_2_O_4_ nanoparticle using a chemical co-precipitation technique. We found in a previous study that the use of NaOH as the co-precipitating agent results in larger particle size for which it is impossible to form stable colloidal suspensions (not published). Therefore, we used NH_4_OH as the co-precipitating agent. Since Mn^2+^ possesses an unstable valence state, we synthesized MnFe_2_O_4_ nanoparticle at room temperature and varied the nanoparticle sizes with pH variations without applying any heat. Thus, we examined the size-dependence of SLP and efficacy of MnFe_2_O_4_ nanoparticle as a negative/positive MRI contrast agent in the rat model.

## 2. Materials and Methods

### 2.1. Sample Preparation

We synthesized Manganese ferrites (MnFe_2_O_4_) nanoparticles using a chemical co-precipitation method for which we used the starting materials (MnCl_2_·4H_2_O and FeCl_3_) of analytical grade and NH_4_OH as the co-precipitating agent. The MnCl_2_·4H_2_O and FeCl_3_ salts were dissolved in distilled water in the required molar ratio of 1:2 and underwent thorough mixing. Then 8M of NH_4_OH solution were added drop-wise by micropipette (H17662, VWR, Radnor, PA, USA) into the above salts’ solutions under continuous stirring by magnetic stirrer (SP250, Lab Depot, Dawsonville, GA, USA). Extra NH_4_OH (6M) was added to maintain the pH to the desired level of 9–12 that plays a determining role in controlling the precipitation and the precipitated particles’ size. The precipitates collected through centrifugation (2–16 P, Sigma, Harz, Germany) at 13,000 rpm for 20 min were washed ten times by centrifugations. The silver nitrate test confirmed that the sample was free from NH_4_OH. The product was then dried in an oven at 80 °C for 72 h for perfect ferritization. The as-dried powder was ground with an agate mortar and pestle to obtain the as-dried MnFe_2_O_4_ nanoparticles. The precipitates of the MnFe_2_O_4_ nanoparticles were obtained according to the following reaction,
MnCl_2_·4H_2_O + 2FeCl_3_ + 8NH_4_OH → MnFe_2_O_4_ + 8NH_4_Cl + 8H_2_O

The aqueous chitosan solution (Sigma-Aldrich, St. Louis, MI, USA) with low molecular weight, 75–80% deacetylated, and viscosity 20 cps (1% solution in 1% acetic acid; Brookfield, Middleboro, MA, USA) was prepared using acetic acid. We mixed 0.40 g chitosan into 40 mL distilled water with a magnetic stirrer at 500 rpm. To form a homogeneous solution, we added 1 mL (2 N) acetic acid drop-wise and stirred until chitosan was fully dissolved in water. Then, the chitosan solution was centrifuged twice at 13,000 rpm for 20 min. The chitosan solution was formulated for coating and surface modification of nanoparticles.

### 2.2. Characterization

Structural characterization of MnFe_2_O_4_ nanoparticles at different particle sizes was performed by a powder X-ray diffractometer (XRD) (PW3040, X’Pert Pro, Philips, Amsterdam, The Netherland) using Rigaku CuK_α_ radiation source in the 2θ ranges from 15° to 75° at 40 kV, 30 mA. The crystalline nature and coating condition were observed by Fourier transform infrared spectroscopy (FTIR) (L1600300 Spectrum TWO UTA ETHERNET, Perkinelmer, Shelton, CT, USA). We analyzed the shape and microstructure of samples using a high-resolution transmission electron microscope (TEM) (F200X Talos, Thermo Fisher, Waltham, MA, USA) at the operating voltage of 200 kV and energy-dispersive X-ray spectroscopy (EDX) (S50 QLD9111, FEI, Amsterdam, The Netherlands). The magnetic state of the samples was determined by the physical properties measurement system (PPMS) (D235, Quantum Design, San Diego, CA, USA) at 5 and 300K using a 5 Tesla magnetic field and also using Mössbauer spectroscopy (MS4, Vincent, Belgrade, Serbia). The hydrodynamic size, polydispersity index (PDI), and zeta potential of chitosan-coated samples at different particle sizes were investigated using dynamic light scattering (DLS) (ZEN3600 Zeta Potential Instrument, Malvern, UK) and zeta potential or electrophoretic mobility technique. The time-dependent temperature profiles of the chitosan-coated samples at different particle sizes were carried out at different concentrations of 1, 2, 3, and 4 mg/mL using the hyperthermia set-up (EASYHEAT 5060LI, Ambrell, Rochester, NY, USA) with a radio-frequency (RF) induction coil of 4 mm diameter and 8 turns with an alternating current (AC) magnetic field of 20 mT at a resonance frequency (342 kHz). The temperature rise of the samples was measured using an optical fiber thermometer. The cytotoxicity assay of chitosan-coated MnFe_2_O_4_ nanoparticles at 2 mg/mL was carried out using live/dead cell assay. HeLa cell line was cultured using a BioSafety cabinet (NU-400-E, NuAire, Plymouth, MN, USA) and a 37 °C + 5% CO_2_ incubator (NuAire, Plymouth, MN, USA), trinocular microscope with a camera of Hemocytometer (Optika, Ponteranica, Italy). Nanoparticle size-dependent of T_2_ relaxivities were determined on phantoms composed of a small tube that contained chitosan-coated MnFe_2_O_4_ at five different concentrations (e.g., 0.17, 0.34, 0.51, 0.68, and 1.03 mM) in four sets of a larger tube of four different nanoparticle size. MRI in vivo studies were carried out in a rat model by a horizontal-bore 7T MRI scanner (MRS7017, MR Solution, Guildford, UK).

### 2.3. Animal Handling and In-Vivo MRI

We performed animal handling following The Guide for the Care and Use of Laboratory Animals (1996) and based on the ARRIVE Guidelines for reporting animal research [52]. We minimized the sufferings of the experimental animal according to the requirement of the Ethical Review Committee of Animal Experiments of Atomic Energy Centre Dhaka, which approved the protocol with Memo No: AECD/ROD/EC/20/201.

Albino Wister rats (male, age 11–12 weeks, weight 190–200 g) received an intraperitoneal injection of ketamine/xylazine (0.5 mg/kg and 10 mg/kg) to sedate (i.e., complete loss of reflexes) them before the MRI scan. Rats received a single dose of CS-coated MnFe_2_O_4_ (10 mg/kg) using a 26G needle in the tail vein. The CS-coated MnFe_2_O_4_ nanoparticles suspended in phosphate buffered solution had a concentration of 2 mg/mL.

The rat was placed on an imaging bed of 66 mm diameter and 426 mm in length that was graduated along its axis to ensure positioning reproducibility. The bedding was devised with a tubular structure allowing warmed up air circulation and a tooth bar to immobilize the animal. There was provision for physiological monitoring and gating. The rat’s head was put inside a transmit/receive RF head coil, which is a quadrature birdcage of inner diameter 65 mm and length 70 mm. The conveyor mechanism transported the entire assembly inside the homogeneous region of the magnetic field of the 7T MRI scanner.

The Carr–Purcell–Meiboon–Gill (CPMG) pulse sequence was used to determine nanoparticles size-dependent T_2_ relaxivity of CS-coated MnFe_2_O_4_ as a negative contrast dye. This experiment using phantoms represented the relaxivity of water protons (*r*_2_) in presence of per mM of CS-coated nanoparticles as a contrast agent. MRA experiments were carried out to examine the performance of CS-coated MnFe_2_O_4_ nanoparticles as MRA or blood pool/positive contrast agent of sizes 6, 10, and 15 nm in rat brain with and without contrast agents using the time-of-flight (TOF) three-dimensional (3D) sequence and Maximum Intensity Projection (MIP).

## 3. Results and Discussion

### 3.1. X-ray Diffraction (XRD) Analysis

XRD analysis reveals the structural characterization of the MnFe_2_O_4_ nanoparticles for the determination of average particle size and phase of the nanoparticles. The XRD patterns of as-dried MnFe_2_O_4_ at different pH presented in Figure 1a provides clear evidence of the formation of the ferrite phase. Bragg’s reflections indexed as (111), (220), (311), (420), (511), (440), and (620) confirmed the formation of a well-defined single phase cubic spinel structure without any detectable impurity phase belonging to the space group Fd3m shown in Figure 1a. A significant broadening of the XRD peaks indicates that the ferrite particles are of nanometric size. The crystallite size of the samples was determined using the maximum intensity peak (311) by Scherrer’s formula [53]: d=0.9λ/βcosθ, where *β* is the full-width half maxima measured in radians, *θ* is the Bragg’s angle, and *λ* = 1.5418 Å is the wavelength of Cu Kα energy. The interplanar spacing ($d_{ip}$) obtained from the XRD pattern, yielded the lattice parameter using the formula,$\;a=d_{ip}\sqrt{\left(h^{2}+k^{2}+l^{2}\right)}$, where *h*, *k* and *l* are the Miller indices. The $d_{ip}$ values and intensities of diffraction peaks matched with the single crystalline MnFe_2_O_4_ (JCPDS Card No. 074–2403). The XRD results of MnFe_2_O_4_ at different pH shown in Figure 1 are comparable with the previously reported results [54,55].

Figure 1b shows the pH dependence of nanoparticle size (d) and lattice parameters (a) extracted from the XRD patterns comparable with the previously reported values [54,55].

We see in Figure 1b that the particle size increases monotonically with the increase of pH, which is 5, 6, 10, and 15 nm at the pH of 9, 10, 11, and 12, respectively, synthesized in four separate batches. Thus, we varied the nanoparticle size by controlling the pH. There was a steady but non-linear increase of lattice parameters with increased pH, which was 8.43, 8.48, 8.49, and 8.50 Å, respectively. From the experimental lattice parameter and grain size, we determined the X-ray density, *d_x_*, the hopping length *L_A_* and *L_B_* (the distance between the magnetic ions on A and B-sites), and the specific surface area of the particles S [56] using the relations $d_{x}=8M_{w}/Na^{3}$, $L_{A}=a_{exp}\sqrt{3}/4$, $L_{B}=a_{exp}\sqrt{2}/4$, and $S=6/\left(d\times d_{x}\right)$. Figure 1c,d show the variations of$\;d_{x}$, $L_{A}$, $L_{B}$, and S with the crystallite size d. Both $d_{x}$ and S decreased with the increase of nanoparticle size d. The hopping length $L_{B}$ was lower than the $L_{A}$, while both $L_{A}$ and $L_{B}$ slightly increased with d because of the change of cation distributions.

In the bulk condition, MnFe_2_O_4_ is a normal spinel with 80% Mn^2+^ occupying tetrahedral (A) sites, while 20% of Mn^2+^ occupying octahedral (B) sites [57]. However, an inversion factor of x changes for the MnFe_2_O_4_ nanoparticles. Barik et al. [58] demonstrated that the change in the degree of inversions (x) with the lattice constants and listed theoretical and experimental lattice parameters of several spinel ferrites of different compositions in which the theoretical and experimental lattice parameters of MnFe_2_O_4_ are quite close. Further, O’Neill et al. [59] demonstrated through a histogram the difference of experimental and calculated lattice constants Δa on 66 simple oxide spinels where Δa was small for most of the oxide spinels. Considering a_exp_
≈
*a_th_* we would get an estimation of the inversion parameter x considering other interactions negligible from the relation of $a_{th}=8/3\sqrt{3}\left[\left(\left(r_{A}+R_{O}\right)+\sqrt{3}\left(r_{B}+R_{O}\right)\right)\right]$. In this relation, $r_{A}$, $r_{B}$, and $R_{o}$ are the ionic radii of Mn^2+^, Fe^3+^, and O^2−^, respectively. Considering the random cation distribution (Mn1−xFex)[Mnx2Fe1−x2]2O4, we determined the inversion parameter x as 0.79, 0.44, 0.37, and 0.30 for the nanoparticle size d of 5, 6, 10, and 15, respectively. The cation distributions became (Mn_0.21_Fe_0.79_) [Mn_0.79_Fe_1.21_] O_4_, (Mn_0.56_Fe_0.44_) [Mn_0.44_Fe_1.56_] O_4,_, (Mn_0.63_Fe_0.37_) [Mn_0.37_Fe_1.63_] O_4_, (Mn_0.70_Fe_0.30_) [Mn_0.30_Fe_1.70_] O_4_ for the particle size of 5, 6, 10 and 15 nm, respectively. This was in line with the fact that with the increase of particle size, the occupancy of Mn^2+^ on A-site increased, which was 80% in the bulk state. Figure 2A shows nanoparticle size (d) dependence of ionic radii on tetrahedral and octahedral sites ($r_{A}$ and $r_{B}$) (left axis) and bond length ($\;d_{AL}$ and $\;d_{BL}$) (right axis), oxygen parameters $u^{\bar{4}3m}$ and $u^{\bar{3}m}$ and shared tetrahedral edge ($d_{AE}$), shared octahedral edge ($d_{\mathrm{BE}}$), unshared octahedral edge ($d_{BEU}$). In Figure 2A, the variations of the ionic radii in tetrahedral and octahedral sites $r_{A}$ and $r_{B}$ demonstrate that $r_{A}$ < $r_{B}$, but with the increase of the nanoparticle size $r_{A}$ increases as 0.525, 0.585, 0.597, and 0.609Å, while $r_{B}$ decreases as 0.707, 0.679, 0.674, and 0.668 Å. This was because with the increase of nanoparticle size, the value of x decreased, which led to the transfer of Mn^2+^ from B site to A site progressively. Similar behavior was observed for tetrahedral and octahedral bond length with the change in the nanoparticle size. We determined the tetrahedral bond length and octahedral bond length,$\;d_{AL}=a_{exp}\sqrt{3}\left(u^{\bar{4}3m}-0.25\right)$ and $d_{BL}=\sqrt{a_{exp}\left(3{\left(u^{\bar{4}3m}\right)}^{2}-11/4u^{\bar{4}3m}+43/64\right)}$. O’Neill et al. [59] suggested that since the cations in ferrites have different sizes, therefore, any change in x would define the lattice parameter, a and oxygen parameter u, which have a unique relation, $u^{\bar{4}3m}=\left(r_{A}+R_{o}\right)/\sqrt{3}a_{exp}$. There are two ways of representation for the oxygen parameter$:\;u^{\bar{4}3m}$, when the origin of the unit cell is considered on an A-site cation $\bar{4}3m$ and $u^{\bar{3}m}$ when the origin of the unit cell is at octahedral vacancy $\bar{3}m$, where, $u^{\bar{3}m}=u^{\bar{4}3m}-1/8$. The oxygen parameters of $u^{\bar{4}3m}$ and $u^{\bar{3}m}$ plotted with the variations of particle size in Figure 2A, are in the same order but different in values. For a perfect fcc structure for which the $u_{ideal}^{\bar{4}3m}$ and $u_{ideal}^{\bar{3}m}$ are 0.375 and 0.250, respectively [56]. O’Neill et al. [59] demonstrated a relationship between the Madelung constant M and the oxygen parameter u. There exists a crossover for u=0.2555 between normal and inverse spinel. The structure is normal above 0.2555 and inversely smaller than this value. We obtained $u^{\bar{3}m}$ as 0.2514, 0.2547, 0.2553, and 0.2560 for 5, 6, 10, and 15 nm, respectively. Thus, we saw that for 5 nm, the structure was mostly inverse with x = 0.79. For the size 6 and 10 nm, the structures were mostly disordered with x = 0.44 and 0.37, and for the size of 15 nm, the structure was mostly normal with x = 0.30. We determined shared tetrahedral edge, $d_{AE}=a_{exp}\sqrt{2}\left(2u^{\bar{4}3m}-0.5\right)$, shared octahedral edge, $d_{BE}=a_{exp}\sqrt{2}\left(1-2u^{\bar{4}3m}\right)$ and unshared octahedral edge $d_{BEU}=\;a_{exp}\sqrt{\left(4u^{2}+3u^{\bar{4}3m}+11/16\right)}$ and presented this in Figure 2A. The shared tetrahedral edge increased with the increase in nanocrystallite size, and the shared octahedral edge decreased while the unshared octahedral edge slightly increasesd. This was because of the transfer of larger cation Mn^2+^ on the A-site and the smaller cation Fe^3+^ on the B-site with increased nanocrystallite sizes.

Figure 2b shows the nanoparticle size dependence of interionic distances between cations (Me-Me) (*b*, *c*, *d*, *e*, *f*), between cations and anions (*p*, *q*, *r*, *s*) and bond angles (*θ*_1_, *θ*_2_, *θ*_3_, *θ*_4_, *θ*_5_). The interionic distances between cations b, c, d, e, and f were calculated using the relations $b=\left(a_{\exp}/4\right)\sqrt{2}$, $c=\left(a_{exp}/8\right)\sqrt{11\;}$, $d=\left(a_{exp}/4\right)\sqrt{3}$, $e=\left(3a_{exp}/8\right)\sqrt{3}$, $f=\left(a_{exp}/4\right)\sqrt{3}$ and cations and anions p, q, r, and s were calculated using the relations $p=a_{exp}\left(1/2-u^{\bar{3}m}\right)$, $q=a_{exp}\left(u^{\bar{3}m}-1/8\right)\sqrt{3}$, $r=a_{exp}\left(u^{\bar{3}m}-1/8\right)\sqrt{11}$, $s=a_{exp}/3\left(u^{\bar{3}m}-1/2\right)\sqrt{3}$. The bond angle was calculated using the relations $\theta_{1}=cos^{-1}\left(\left(p^{2}+q^{2}-c^{2}\right)/2pq\right)$, $\theta_{2}=cos^{-1}\left(\left(p^{2}+r^{2}-e^{2}\right)/2pr\right)$, $\theta_{3}=cos^{-1}\left(\left(2p^{2}-b^{2}\right)/2pr\right)$, $\theta_{4}=cos^{-1}\left(\left(p^{2}+s^{2}-f^{2}\right)/2ps\right)$, $\theta_{5}=cos^{-1}\left(\left(r^{2}+q^{2}-d^{2}\right)/2rq\right)$. We found that when nanoparticle size increased, the value of x decreased, and therefore, Mn^2+^ was transferred to the A-site replacing Fe^3+^ to B-site. From Figure 2b, it can be assumed that the magnetization should increase with the increase in nanocrystallite size because of the increase of B-B interactions as the bond length B-O, p decreases and the bond angle B-O-B, $\theta_{3}$ and $\theta_{4}$ increase because of the migration of Mn^2+^ to A-site and Fe^3+^ to B-site with the increase of crystallite size.

### 3.2. TEM and EDX Analysis

Figure 3a–d shows the TEM bright field, dark field, selected area diffraction pattern, and high-resolution TEM images of the CS-coated nanoparticles synthesized at pH = 10 with the crystallite size of 6 nm obtained from XRD. Figure 3a shows bright-field images acquired from the transmission beam, while Figure 3b shows dark field images acquired by a diffracted beam. Selected area diffraction (SAED) patterns in Figure 3c provide information on the nanoparticles’ structure and the high-resolutoin TEM image presented in Figure 3d demonstrates crystallinity. The d-values of the diffraction rings in Figure 3c were determined using Velox software and yielded values that exactly match the literature, and the diffraction pattern was indexed accordingly. The SAED pattern was consistent with the XRD pattern showing the cubic spinel structure of the MnFe_2_O_4_ nanoparticles belonging to the Fd3m space group.

From the diffraction circle, conical dark-field images acquired using Velox software. Both the bright field and dark field images show the dispersion of the coated nanoparticles. The high-resolutoin TEM image in Figure 3d along the zone axis indicates good crystallinity. The d-spacing depicted from the high-resolutoin TEM image was d_311_ = 251.7 pm.

The EDX technique provides an effective atomic concentration of the sample on top surface layers of the solids under investigations. The EDX spectrum of MnFe_2_O_4_ nanoparticle in as-dried condition at room temperature shows the peaks of Mn, Fe, and O along with the C substrate peak. The microanalysis of EDX data indicates the constituent elements as Mn (at% 12.33) Fe (at% 31.53), and O (at% 36.65), respectively.

### 3.3. Magnetic Measurements

Figure 4a–d shows M-H hysteresis loops of the bare MnFe_2_O_4_ nanoparticle of sizes of 5, 6, 10, and 15 nm measured at 5 and 300 K with a maximum applied magnetic field, H_max_ = 5 Tesla. The negligible coercivity indicated that the MnFe_2_O_4_ nanoparticles of different particle sizes exhibited a typical superparamagnetic nature [56] with small interactions between the particles. A small volume fraction of ferromagnetic phases was embedded in the superparamagnetic matrix. The increase of all the parameters with the nanoparticle size, such as maximum magnetization (M_max_), coercivity (H_c_), and remnant ratio (M_r_/M_max_) indicated that inter-particle interactions increased with the increase of nanoparticle size. Two types of interactions exist in the magnetic nanoparticle, which are exchange and dipolar interactions. For sufficiently small particle size, thermal energy dominates over exchange energy, which reduces M_max_, H_c_, M_r_/M_max_.

Further demagnetization occurs in superparamagnetic nanoparticles by the dipolar interactions, which reduces all above parameters [60]. At 5 K, however, exchange energy dominates over thermal energy for which exchange energy overcomes the barrier of thermal energy. The dipolar interactions weaken as a result of which M_max_, H_c_, and M_r_/M_max_ increase, and the samples show ferrimagnetic nature. M-H curves indicate that at 300 K with an applied magnetic field of 5 T, the magnetization is not saturated. Maximum magnetization recorded from the M-H curve was 11, 25, 34, and 66 emu/g with an applied field of 5 Tesla for the particle sizes of 5, 6, 10, and 15 nm, respectively at 5 K, whereas maximum magnetization values were 8, 12, 19 and 41 emu/g at 300 K. These values were lower than that of the bulk manganese ferrite, which was 80 emu/g, reported in the literature [61,62]. The smaller quantity of magnetization may be due to the higher percentages of atoms located on the nanoparticles’ surface, producing a magnetically inactive layer or disordered layer on the surface.

Figure 5 shows the variations of M_max_, magnetic moment (n_B_), H_c_, and M_r_/M_max_ with particle sizes at 5 and 300 K acquired from M-H loops. The magnetic phase transition from ferromagnetic to superparamagnetic state occurred between these two temperatures of 5 and 300 K. In Figure 5a, there is an increase of M_max_ from 11 to 66 emu/gm at 5 K and 8 to 41 emu/gm at 300 K with the increase of particle size from 5 to 15 nm. Both M_r_/M_max_ and H_c_ decreased significantly from 5 K to 300 K because of the ferrimagnetic to superparamagnetic transition. Using the formula, $n_{B}^{e}=\frac{M_{s}\;\times \;M_{w}}{5585}\;Bohr\;magneton/f.u$ gives an experimental magnetic moment $n_{B}^{e}$. Figure 5b shows the variations of $n_{B}^{e}$ with particle size. The theoretical magnetic moment of MnFe_2_O_4_ using Néel’s two sublattice model was $5\mu_{B}$ irrespective of cation distribution for MnFe_2_O_4_ because of the similar magnetic moment of $5\mu_{B}$ for both Mn^2+^ and Fe^3+^. The values of $n_{B}^{e}$ for the particle sizes of 5, 6, 10, and 15 nm were 0.46, 1.05, 1.43, and 2.77$\mu_{B}$ at 5 K, whereas, 0.34, 0.50, 0.79, and 1.72 $\mu_{B}$ at 300 K. The differences between the theoretical and experimental magnetic moment showed that we need to invoke Yafet–Kittel’s three sublattice model [63]. We determined the Canting angles using the relation, $\alpha_{YK}=cos^{-1}\left(\frac{n_{B}^{e}+M_{A}}{M_{B}}\right)$, where, $M_{A}$ and $M_{B}$ are the magnetic moments on A and B sites. Figure 5c presents the nanoparticle size-dependence of canting angles, which shows almost a linear relationship. The $\alpha_{YK}$ at both 5 and 300 K decreases with the increase of nanoparticle size from 5 to 15 nm.

Consequently, we see a linear increase of magnetic moment and magnetization with an increase in nanoparticle diameter. Figure 5d–f presents the variations of effective anisotropy, squareness ratio, and coercivity with the size of nanoparticles. A large enhancement in anisotropy at 5 K than 300 K demonstrates that the nanoparticles are in the blocked state at 5 K, whereas at 300 K the nanoparticles are in the superparamagnetic state.

This change manifests in the variations of M_r_/M_max_ and H_c_. The M_r_/M_max_ changed from 0.27 to 0.38 at 5 K, and 0.038 to 0.045 at 300 K with the increase of the nanoparticle size, and the H_c_ increased from 76 to 93 Oe at 5 K and from 4 to 10 Oe at 300 K.

We analyzed magnetization data with an inversion parameter *x*, oxygen parameter $u^{\bar{4}3m}$, bond length *p*, and *q* and the bond angle $\theta_{2}$ and $\theta_{3}$. The results presented in Figure 6a show that $u^{\bar{4}3m}$ decreases linearly with the increase of x, i.e., with the increase of Mn^2+^ on the A-site as a result of which lattice parameter increases and also increases the oxygen parameter. MnFe_2_O_4_ is essentially a normal spinel in the bulk state with 20% Mn^2+^ on the B-site. We see that with the increase of nanoparticle size, the concentration of Mn^2+^ ions on the B-site decreases, which reduces the distortion of the MnFe_2_O_4_ spinel structure. In Figure 6a, magnetization increases with the decrease in the inversion parameter, *x*, i.e., with the decrease in Mn^2+^ ions on B-site and with subsequent increase in Fe^3+^ ions on B-site shown in Figure 6b. The increase in Fe^3+^ ions on the B-site oxygen parameter increases as a result of which magnetization increases, as shown in Figure 6c. We already found from Figure 2B that the nanoparticle size dependence of bond length, *p* decreases, and the bond angles $\theta_{3}$ and $\theta_{4}$ increase. Magnetization increases because of an increase in the B-O-B bond angles, $\theta_{3}$ and $\theta_{4}$, and reduction of B-O bond length, *p*, because of the increase in B-B interactions.

### 3.4. Mössbauer Spectroscopy Analysis

Mössbauer spectroscopy is a tool to probe the hyperfine parameter of magnetic nanoparticles. Figure 7a–d represents the Mössbauer spectroscopy of MnFe_2_O_4_ nanoparticles for nanoparticle size of 5, 6, 10, and 15 nm. Table 1 presents the hyperfine parameters such as chemical shift, quadrupole splitting, hyperfine magnetic field, and relative area extracted from the experimental and theoretical fitting of the Mössbauer spectra. We observed from Figure 7 that the samples of different sizes consist of a central doublet region, which confirmed that the major magnetic phase of the samples for all the nanoparticle sizes was largely superparamagnetic. From Table 1, we observe that three subspecies were required to fit the experimental data. Two of them exhibited fast relaxation and represented superparamagnetic phases, and the other one showed slow relaxation that represents ferrimagnetic phases. The area of the ferrimagnetic phase increased with the increase of nanoparticle size, which was 10, 15, 25, and 25% for the size of 5, 6, 10, and 15 nm. Thus, the volume fraction of the ferromagnetic phase increased with the increase of nanoparticle size. The quadrupole splitting, ΔE_q_ of Fe^2+^ was about 3 mm/s, which was much larger than Fe^3+^, and the ΔE_q_ values of a low spin of Fe^3+^ were smaller than 0.8 mm/s. The ΔE_q_ values presented in Table 1 indicate that iron is present in the form of Fe^3+^ and low spin [64]. It is critical to estimate cation distributions because most of the Fe^3+^ are magnetically isolated with the nonmagnetic coordination atoms, and they do not contribute to the long-range magnetic order [64].

### 3.5. Fourier Transform Infrared Spectroscopy (FTIR) Analysis

Figure 8a–d presents the FTIR spectra of uncoated, CS-coated MnFe_2_O_4_ nanoparticles and pure CS having different nanoparticle sizes. For the uncoated sample, two peaks were found near the wave number 400 cm^−1^ and 600 cm^−1^, due to metal- oxide stretching bonds at tetrahedral sites and octahedral sites. Another peak was found near the wave number 3500 cm^−1^, which was due to the O-H stretching band of associated water bound with the sample in the as-dried condition. For pure CS, peaks were found at waves number 1090, 1420, 1610, and 2850 cm^−1^ due to the stretching vibration of C-O-C- in a glycosidic linkage, CH_3_ in amide group, NH_2_ in the amino group, and CH_2_ stretching vibration to a pyranose ring which was a characteristic peak of chitosan. CS-coated MnFe_2_O_4_ samples have similar peaks of CS, and the peaks are shifted, which means the samples coated well with CS [65]. We observed the similarity in behaviors for other particle sizes, i.e., for the nanoparticle sizes of 6, 10, and 15 nm.

### 3.6. Dynamic Light Scattering (DLS) Measurements

We carried out DLS measures to determine the hydrodynamic size (H_d_) and polydispersity index (PDI). The hydrodynamic size (H_d_) is the size of the MNPs in association with the hydration layer around the nanoparticle, and the polydispersity index (PDI) indicates the degree of dispersion of nanoparticle in a colloidal suspension. Figure 9a,b represents the concentration and size dependence of the hydrodynamic diameter distribution (H_d_). Figure 9c,d presents the mean H_d_ and PDI of CS-coated MnFe_2_O_4_ nanoparticles with concentration and nanoparticle sizes.

In the DLS measurement, laser beam incident on the nanoparticle moves with dragging force to modify their surfaces due to coating elements and hydration layer, which causes their sizes to increase from 5 to 100 nm [66]. The average H_d_ was found to vary from 86 to 149 nm for different concentrations of nanoparticles and from 94 to 150 nm for different nanoparticle sizes, whereas the average core size was in the range of 5–15 nm from XRD. Therefore, H_d_ size distributions are greater than the core diameter observed by TEM. Demirci et al. [67] found the particle agglomerations in size range ~60–300 nm for MnFe_2_O_4_ nanoparticles. For biomedical applications, hydrodynamic diameter and the PDI are critical parameters. The PDI determines the extent of aggregation, and a lower PDI value is a prerequisite for biomedical applications. The range of hydrodynamic diameter for the biomedical applications should be less than 250 nm with a PDI value of less than 0.300 [68]. The H_d_ of the CS-coated MnFe_2_O_4_ nanoparticles in this study was less than 250 nm, and the PDI was nearly 0.300, which were satisfactory for biomedical applications.

### 3.7. Zeta Potential

We studied zeta potentials of CS-coated MnFe_2_O_4_ nanoparticles across sizes. Nanoparticles have a surface charge that attracts a thin layer of ions of opposite charge to the nanoparticle surface. The double layers of ions travel with the nanoparticle as it diffuses throughout the solution. The electric potential at the boundary of the double layer is known as the zeta potential of the particles and has values that typically range from +100 mV to −100 mV. Zeta potential is a tool for understanding the stability and coating condition of the nanoparticle in the colloidal suspension. The surface charge may be positive or negative, depending on the solution’s nature by generating ionizable functional groups of nanoparticles [69].

Zeta potential increases with rising electrophoretic, electrostatic, hydrophilic material, and hydrophobic organic surface charge mobility by changing the (+ve), (−ve) charge of nanoparticles in disperse solution. If the suspension is stable, it means that the suspension possesses a high zeta potential value. The magnitude of the zeta potential is a determining factor of colloidal stability, which nominally holds values greater than +30 mV or less than −30 mV for stable colloidal suspension [70,71]. Dispersions with a low zeta potential will eventually aggregate due to Van der Waals’s interparticle attractions. Figure 10 presents the distribution of the zeta potential for CS-coated MnFe_2_O_4_ of the concentration of 2 mg/mL. We see from Figure 10 that the value of zeta potential for CS-coated MnFe_2_O_4_ nanoparticles at 2 mg/mL concentration is +47 mV, +46 mV, +44 mV, and +41 mV for the nanoparticle sizes of 5, 6, 10, and 15 nm respectively, which are higher than +30 mV and is satisfactory for stable colloidal suspension.

### 3.8. Cytotoxicity Analysis

Cytotoxicity is crucial for biomedical applications. We examined the cytotoxicity of CS-coated MnFe_2_O_4_ nanoparticles in HeLa cells cultured to a confluent state in DMEM (Dulbecco’s Modified Eagles’ Medium) containing 1% penicillin-streptomycin, gentamycin, and 10% fetal bovine serum (FBS). Cells (4 × 10^5^/200 µL) were seeded onto 48 well plates and incubated at 37 °C + 5% CO_2_. After 24 h, we added 25 µL of the sample (filtered) in each well. We examined cell mortality under an inverted light microscope after 24 h of incubation. After 24 h of incubation, insoluble samples washed out with fresh media. We then examined the cytotoxicity using a Hemocytometer and inverted light trinocular microscope. We repeated this examination twice. Figure 11a,b shows the images of the HeLa cells, which reflect the cytotoxic effect of CS-coated MnFe_2_O_4_ samples with the particle sizes of 5, 6, 10, and 15 nm at the concentration of 2 mg/mL, as well as the medium using a solvent and without solvent as a control. By this figure, it is clear that the cells’ survival was 100% and 95% in the absence and presence of the solvent, respectively.

On the other hand, to observe the effect of toxicity of the coated particles with biocompatible polymer materials on Hela cell lines, four different solutions with different particle sizes were prepared, and the toxicity effect was observed using a hemocytometer. The survival cells were 90%, 90%, 90%, and 85% observed for the particle sizes of 5, 6, 10, and 15 nm, respectively, as shown in Figure 11b. Therefore, no cytotoxic effect was observed for the chitosan-coated MnFe_2_O_4._ Thus, the prepared manganese ferrite nanoparticles were nontoxic. Almost every cell survived when incubated with the sample, which meant the sample itself was nontoxic. The mortality of the cells will not occur when this sample is applied to the body for localized hyperthermia and as an MRI contrast dye. These nanoparticles are, therefore, noncytotoxic and viable for cell lines. These culture studies should be taken in the context of the doses used in the in vivo MRI analysis, but future in vivo cytotoxicity tests would further establish the lethal doses for these nanomaterials.

### 3.9. Magnetic Hyperthermia with Specific Loss Power (SLP)

Hyperthermia is a minimally invasive therapeutic technique for the selective heat treatment approach in cancer therapy, justified by the cancerous cell’s vulnerability to high temperatures. Cancer cells have a high potential to be destroyed at about 42 °C while normal cells can survive at temperatures higher than 46 °C, which offers a window of hyperthermia therapy [72]. Magnetic particle hyperthermia is based on the magnetic nanoparticles as heat mediators when subjected to an alternating magnetic field.

Figure 12a–d shows the time-dependent temperature rise of chitosan-coated MnFe_2_O_4_ nanoparticles of different concentrations with different particle sizes. The heat was applied with a radio frequency magnetic field with an amplitude of 20 mT and a resonance frequency of 342 kHz. The sample was placed inside an induction coil attached to a heating station, and the power controlled by the power supply. The RF current switched on and off for different holding times. An optical fiber thermometer recorded the temperature. From Figure 12a–d, it is clear that the temperature increases with the increase of particle sizes and the concentrations of MnFe_2_O_4_ nanoparticles. The synthesized nanoparticles with particle sizes of 6, 10, and 15 nm for different concentrations reached threshold temperatures (42 °C). Figure 12d represents the maximum temperature (T_max_) rise of chitosan-coated MnFe_2_O_4_ nanoparticle solutions with the nanoparticle size, d. The rise of temperature of the chitosan-coated MnFe_2_O_4_ nanoparticle in an AC magnetic field could be attributed to the different processes of magnetization reversal such as magnetic hysteresis, Néel and Brownian relaxation, and eddy current losses of the magnetic nanoparticles activated by the AC magnetic field [9,13,73,74,75,76]. Eddy current losses are negligible due to the high resistivity of ferrites. In the current study, MnFe_2_O_4_ nanoparticles showed superparamagnetic behavior, and therefore, heat generation due to hysteresis loss partially contributed to the total heat. Néel rotations caused by the spins of the nanoparticles and Brownian rotations resulted from the particle movement to align itself with the magnetic field contribute to the total amount of heat generation. The required therapeutic temperature of the samples having particle sizes of 6, 10, and 15 nm was reached within 20 min for 2, 3, and 4 mg/mL sample concentrations. For 4 mg/mL concentration, the time required to reach the desired hyperthermia temperature was less than the 2 and 3 mg/mL concentrations. This is because there are more magnetic particles for 4 mg/mL concentration resulting in an increased particle-particle interaction, which increases the exchange coupling energy and affects the heating characteristics^33^. The plateau temperature of the time-dependent temperature curve over a prolonged period opens up the possibility of using nanoparticle heating for drug delivery, hyperthermia treatment of cancer, and other targeted therapy. The T_max_ for all samples with different concentrations and different particle sizes shown in Figure 12e shows that the temperature increases with particle sizes. The efficiency of the heating capacity of a magnetic material is quantified through SLP, defined as the amount of heat generated per unit gram of magnetic material per unit time. The specific loss power from the time dependence temperature profile of each curve was determined from the slope of the linear rise of temperature with time using the formula: SLP=C/m × ΔT/Δt, where *C* is the specific heat of water, ΔT/Δt is the linear rise of temperature with time, and m is the mass of the nanoparticles of solution. The SLP value should be as high as possible to minimize the number of magnetic nanoparticles applied for hyperthermia. It is interesting to note that though there is a temperature rise in the case of higher sample concentration, the values of SLP are higher for lower sample concentration and particle sizes shown in Figure 12f. This may be attributed to high particle concentration results in particle agglomeration, suggesting stronger dipole–dipole interactions, which decrease magnetic heating efficiency [77]. Similar results were observed by Urtizberea et al. [78], Presa et al. [79], and Martinez-Boubeta et al. [80] who found that SLP decreases with increasing concentration of iron-oxide nanoparticles. Smaller nanoparticle size gives rise to higher Néel and Brownian motion. Further, smaller particle size gives rise to a smaller hysteresis area. Again, at lower concentrations, Néel and Brownian motion and the hysteresis area are smaller, giving rise to a higher specific loss. Concentration optimization of the nanoparticle is crucial for biomedical applications.

### 3.10. MRI Analysis

#### 3.10.1. MnFe_2_O_4_ as Negative Contrast Agent

MRI is a diagnostic technique widely employed due to its ability to distinguish between healthy and pathological tissues. Manganese ferrites are good T_2_ contrast agents of MRI though, at exceedingly small nanoparticle size, MnFe_2_O_4_ nanoparticles exhibit good positive blood pool contrast [81,82]. As a T_2_ contrast agent, the parameter which determines the quality of contrast is known as relaxivity (*r*_2_). To determine *r_2_* relaxivity in this study, a series of phantom images were acquired at a different echo time. Phantom images were acquired for MnFe_2_O_4_ nanoparticle solutions at different particle sizes. We developed phantoms by filling small Eppendorf tubes with five different concentrations for each particle size. The Eppendorf tubes were then inserted inside a 50 mL falcon tube, which was placed inside a mouse body coil. The CMPG pulse sequence was used to determine T_2_ relaxation in which the repetition time T_R_ was 4000 ms. The series of images procured at echo times (TEs) are 7, 14, 21, 28, 35, 42, 49, 56, 63, 70, 77, 84, 91, 98, 105, and 112 ms.

Figure 13a represents a slice at TE of 14 ms for the particle sizes of 5, 6, 10, and 15 nm. The degree of darkening increases with the increase of the concentration of the nanoparticle. We recorded each voxel’s intensity at different echo times, which dropped exponentially with the increase of echo time. The relaxation time of each voxel determined from the exponential relation of intensity drop with the relaxation time: $I=I_{o}\;exp\left(\left(-\tau \right)/T_{2}\;\right)$, where *I_0_* is the maximum intensity, *T*_2_ is echo time, and τ is relaxation time. The inverse of relaxation time $1/T_{2}\;$is also known as relaxation. In Figure 13b, concentration dependence of relaxation at different particle size = 5, 6, 10, 15 nm are presented.

The variation of relaxation with concentration is fully linear. From the slope of the linear fitting, we determined relaxivity r_2_ for each particle size. The r_2_ relaxivities were 84 (±08) mM^−1^s^−1^, 90 (±07) mM^−1^s^−1^, 103 (±06) mM^−1^s^−1,^ and 125 (±13) mM^−1^s^−1^ for the nanoparticle sizes of 5, 6, 10, and 15 nm, respectively. The relaxivity (*r_2_*) or relaxation (*R_2_*) is directly related to the magnetic moment and particle sizes of nanoparticles [83]. Therefore, particles of higher magnetic moments, smaller particle sizes produce shorter relaxation time, which is essential for good negative contrast enhancement. Figure 13c shows linear dependence of nanoparticle size dependence of *r*_2_ relaxivity.

#### 3.10.2. Magnetic Resonance Angiography (MRA) with Time-of-Flight (TOF)

We tested the contrast efficacy of CS-coated MnFe_2_O_4_ nanoparticles for the blood pool imaging or MRA experiment using the TOF 3D sequence. Figure 14a shows image slices with and without contrast agents for 5 nm, while Figure 14b represents images for 10 nm, and Figure 14c shows the image slices for the size of 15 nm. To acquire the angiography images, we used the repetition time (TR) of 22 milliseconds, echo time (TE) of 3 ms, with a 30° flip angle. The total numbers of scans were 168, with a thickness of 2 mm and FOV of 36 × 36. The reconstruction of the images was achieved using Maximum Intensity Projection (MIP) post-processing software. For the nanoparticle size of 6 and 10 nm in Figure 14a,b the contrast agents enhanced the contrast of the rat veins than the veins of pre-injection control, while for the nanoparticle size of 15 nm, the contrast of the image deteriorated in the post-injection image than the pre-injection control in Figure 14c. Previous studies [82,83] demonstrated that Mn-based contrast agents possess similar characteristics as Gd-based contrast agents that enhance T_1_ or positive contrast of MRI. MnFe_2_O_4_ is the most nontoxic form among other Mn-based compounds. Previous studies [84,85] demonstrated that to act as a positive contrast agent, r_2_/r_1_ should be minimum where r_1,2_ are the relaxivities of contrast agents for which nanoparticle size should be exceedingly small, preferably less than 5 nm. In this study, we see in Figure 14a,b the pre- (A and B) and post-contrast agent injecting images (A’ and B’) that contrast-enhancing efficacy of chitosan-coated MnFe_2_O_4_ as the MRA contrast agent is higher when the nanoparticle size is 5 and 10 nm. With the increase of nanoparticle size as 15 nm in Figure 14c, the contrast-enhancing efficacy of the MnFe_2_O_4_ nanoparticles degrades as evidenced by comparing the images of pre- (A and B) and post-contrast agent (A’ and B’) injecting images.

## 4. Conclusions

Different characterization techniques yielded good coordination to demonstrate that CS-coated MnFe_2_O_4_ nanoparticles of different sizes undergo successful surface modifications. From the hyperthermia studies, we see that with a RF magnetic field of amplitude 26 mT, the SLP in the range of 100–330 depending on the particle sizes and concentrations. The results suggested that to obtain a trade-off between higher specific loss power and T_max_, MnFe_2_O_4_ nanoparticles sizes should be in the range of 6–10 nm and the concentration in the range of 2–3 mg/mL. In vitro and in vivo imaging performed by MRI at 7T shows promising results. T_2_ calculated from the concentration dependence of relaxation and were 84 (±08) mM^−1^s^−1^, 90 (±07) mM^−1^s^−1^, 103 (±06) mM^−1^s^−1^ and 125 (±13) mM^−1^s^−1^ for the nanoparticle sizes of 5, 6, 10, and 15 nm, respectively. The MRA studies are in line with previous studies, which confirm that to use nontoxic, biocompatible, and biodegradable MnFe_2_O_4_ as MRA contrast agents, nanoparticle sizes should be exceedingly small, preferably less than 5 nm. Thus, CS-coated MnFe_2_O_4_ has the promise of using both positive and negative contrast agents of MRI and hyperthermia with nanoparticle size variations.

## Figures and Tables

**Figure 1 nanomaterials-10-02297-f001:**
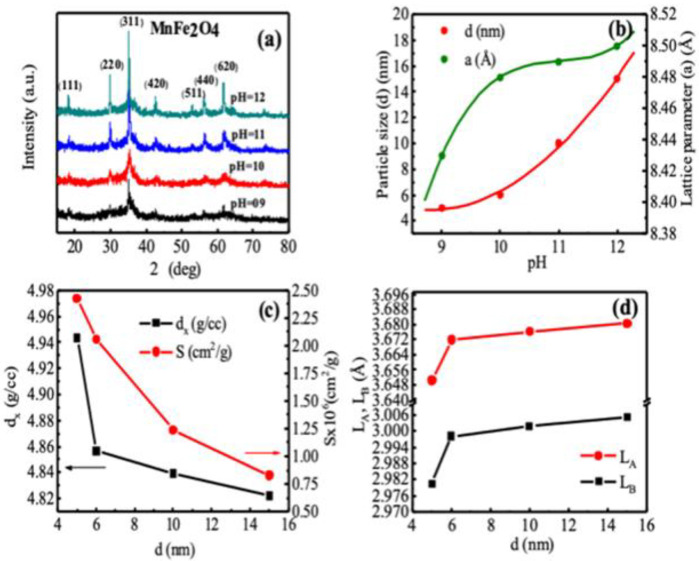
XRD studies of as-dried MnFe_2_O_4_ nanoparticles at different pH. (**a**) XRD patterns of the samples synthesized at pH of 9, 10, 11, and 12, (**b**) pH dependence particle size (d) and lattice parameter (a), (**c**) nanoparticle size dependence of X-ray density (d_x_) and specific surface area (S), (**d**) nanoparticle size dependence of hopping length (L) for tetrahedral (A) and octahedral (B) site.

**Figure 2 nanomaterials-10-02297-f002:**
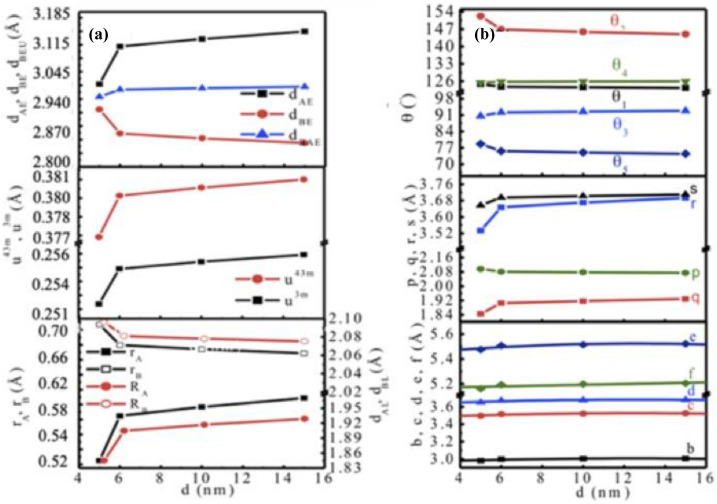
XRD studies of as-dried MnFe_2_O_4_ nanoparticles synthesized at different pH. (**a**) nanoparticle size dependence of ionic radii on tetrahedral and octahedral sites (r_A_ and r_B_) (left axis) and bond length (d_AL_ and d_BL_) (right axis), oxygen parameters (u) and shared tetrahedral edge (d_AE_), shared octahedral edge (d_BE_), unshared octahedral edge (d_BEU_), (**b**) nanoparticle size dependence of interionic distances between cations (Me-Me) (b, c, d, e, f), between cations and anions (*p*, *q*, *r*, *s*) and bond angles (*θ*_1_, *θ*_2_, *θ*_3_, *θ*_4_, *θ*_5_).

**Figure 3 nanomaterials-10-02297-f003:**
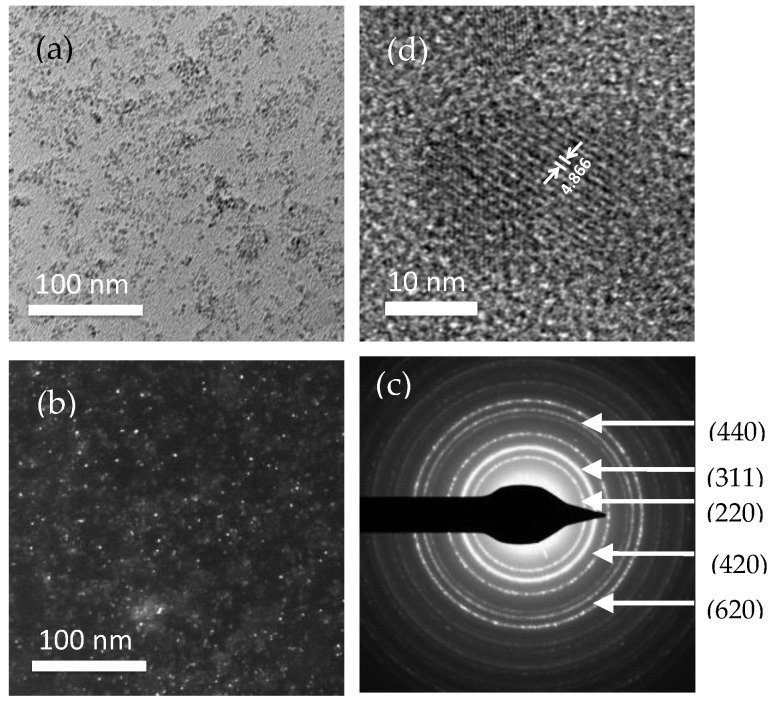
TEM images of the CS-coated MnFe_2_O_4_ nanoparticles at pH 11. (**a**) bright field (BF), (**b**) dark-field (DF), (**c**) selected area diffraction (SAED) pattern, and (**d**) high-resolution TEM image.

**Figure 4 nanomaterials-10-02297-f004:**
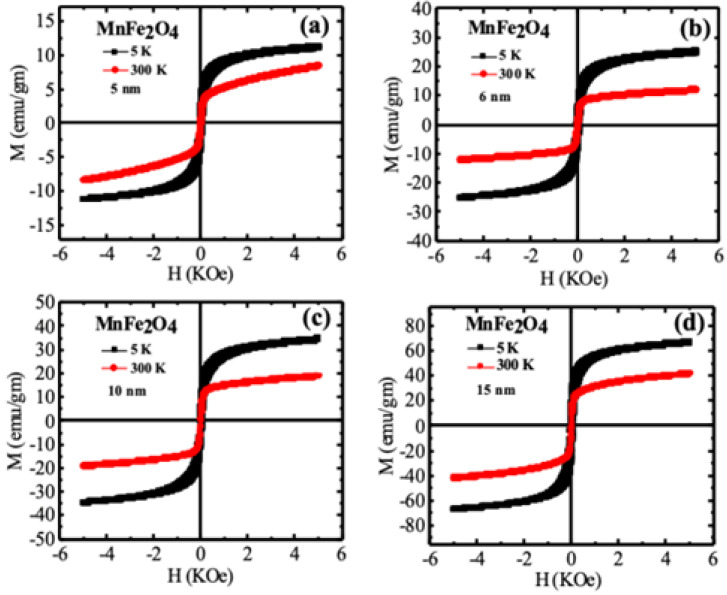
M-H hysteresis loops of MnFe_2_O_4_ nanoparticles in the as-dried condition measured at 5 and 300 K with a maximum applied field of 5 Tesla. The M-H loops of the MnFe_2_O_4_ nanoparticles at different pH are presented, (**a**) 5 nm, (**b**) 6 nm, (**c**) 10 nm, and (**d**) 15 nm.

**Figure 5 nanomaterials-10-02297-f005:**
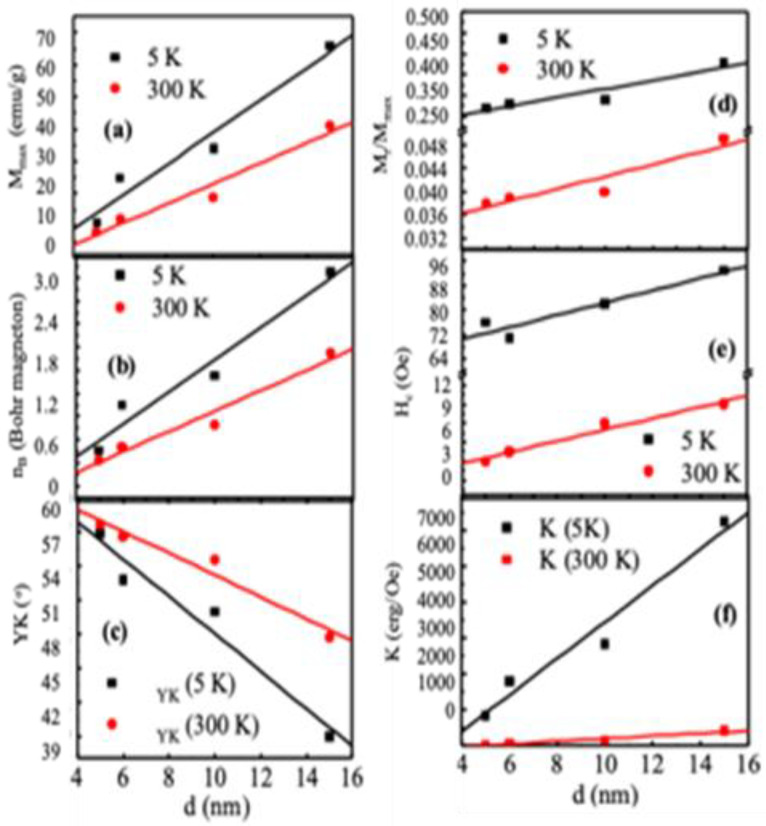
(**a**) Maximum magnetization (**b**) Bohr magneton, (**c**) Canting angles, (**d**) squareness ratio, (**e**) coercivity, and (**f**) anisotropy with size variations of MnFe_2_O_4_ nanoparticles.

**Figure 6 nanomaterials-10-02297-f006:**
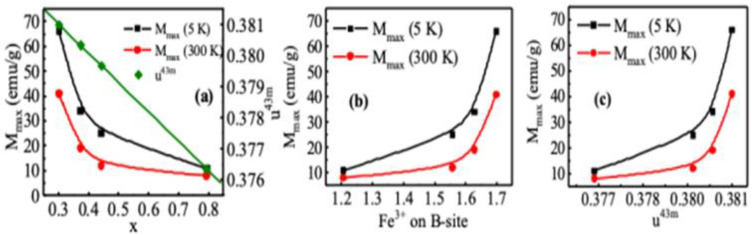
Maximum magnetization with (**a**) inversion parameter, (**b**) with cation distribution in the octahedral site, and (**c**) with oxygen parameter.

**Figure 7 nanomaterials-10-02297-f007:**
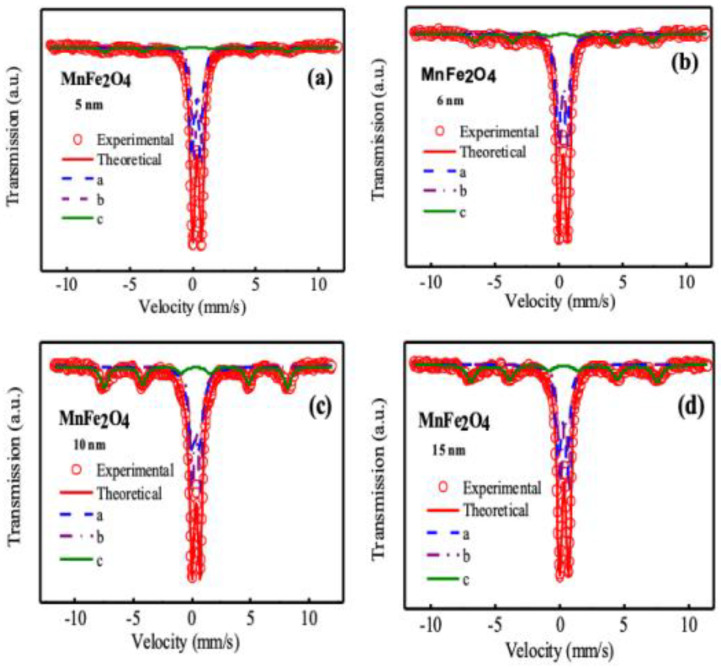
Mössbauer spectra of MnFe_2_O_4_ nanoparticles in the as-dried condition for different nanoparticle sizes measured at room temperature and without any applied magnetic field (**a**) 5 nm, (**b**) 6 nm, (**c**) 10 nm, and (**d**) 15 nm. In the spectrum, hollow red circles represent experimental data, the solid red line represents theoretical fitting, and the other three lines represent sub spectra of three subspecies named a, b, and c to fit the data.

**Figure 8 nanomaterials-10-02297-f008:**
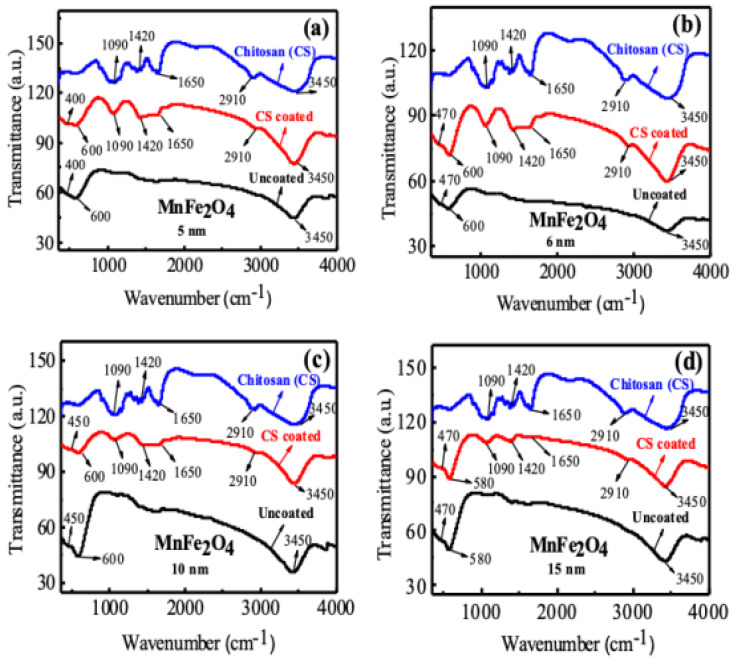
FTIR spectra of uncoated MnFe_2_O_4_, CS-coated MnFe_2_O_4_, and CS nanoparticles for sizes of (**a**) 5 nm, (**b**) 6 nm, (**c**) 10 nm, and (**d**) 15 nm.

**Figure 9 nanomaterials-10-02297-f009:**
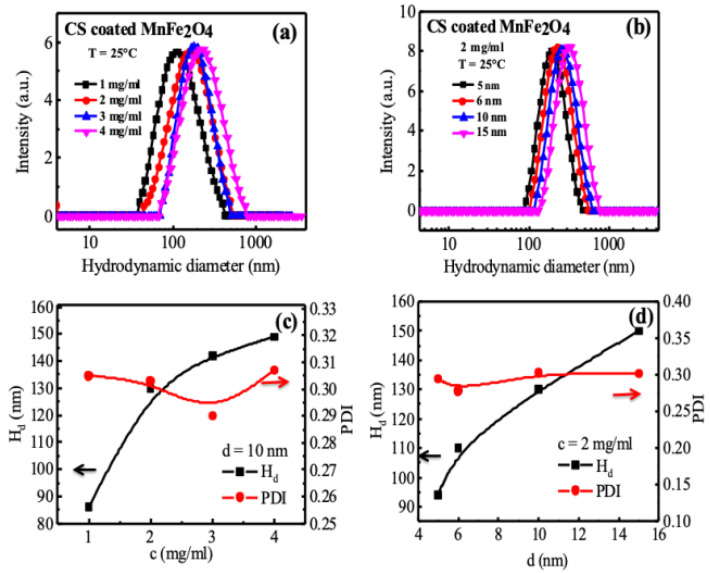
Hydrodynamic diameter (H_d_) of CS-coated MnFe_2_O_4_ nanoparticles measured at 25 °C. In the figure, we present (**a**) distribution of H_d_ with concentration and (**b**) with nanoparticle size at 25 °C. Analyzing the data in (**a**) and (**b**), we presented (**c**) concentration and (**d**) nanoparticle size dependence of hydrodynamic diameter (H_d_) and the polydispersity index (PDI).

**Figure 10 nanomaterials-10-02297-f010:**
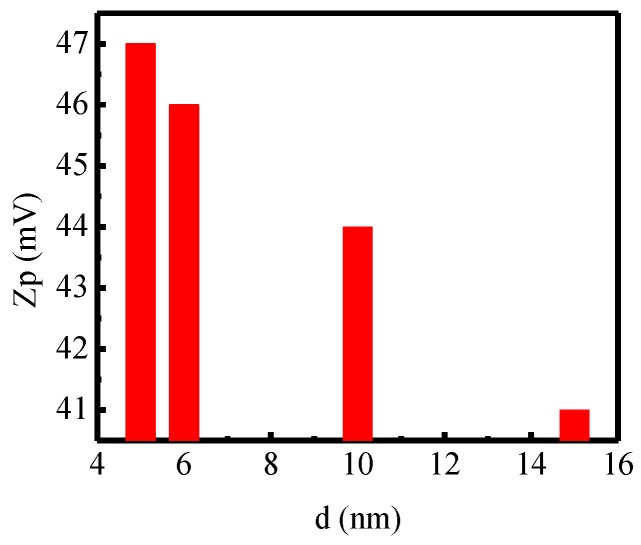
Zeta potential of CS-coated MnFe_2_O_4_ nanoparticles for 5 nm, 6 nm, 10 nm, and 15 nm at room temperature during the DLS measurement.

**Figure 11 nanomaterials-10-02297-f011:**
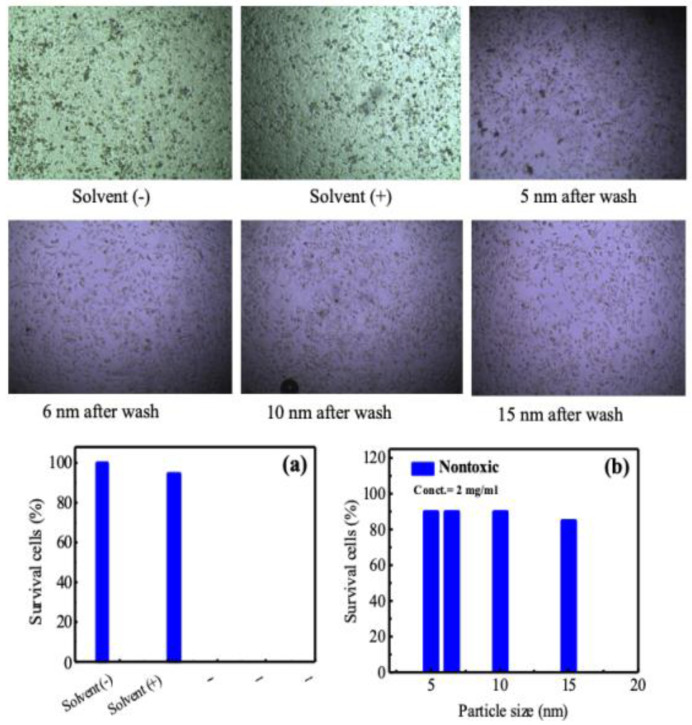
Cytotoxicity results assessed by survival of Hela cells (**a**) with and without solvent and (**b**) CS-coated MnFe_2_O_4_ nanoparticles at different particle sizes, where the corresponding cell culures are pictured on the top panel.

**Figure 12 nanomaterials-10-02297-f012:**
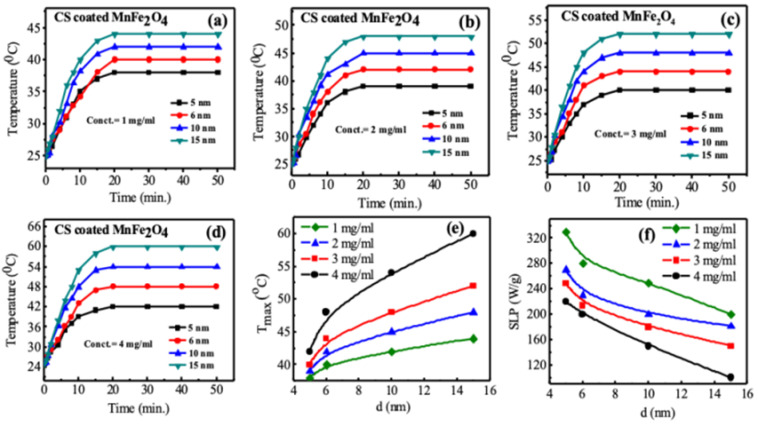
Time dependence temperature curves of CS-coated MnFe_2_O_4_ nanoparticles with a RF magnetic field of an amplitude of 26 mT and a frequency of 342 kHz. The curve presents time dependence of temperature curves of (**a**) 1 mg/mL, (**b**) 2 mg/mL, (**c**) 3 mg/mL, (**d**) 4 mg/mL. Subsequently, nanoparticle size dependence of (**e**) maximum temperature (T_max_), and (**f**) specific loss power (SLP) are presented.

**Figure 13 nanomaterials-10-02297-f013:**
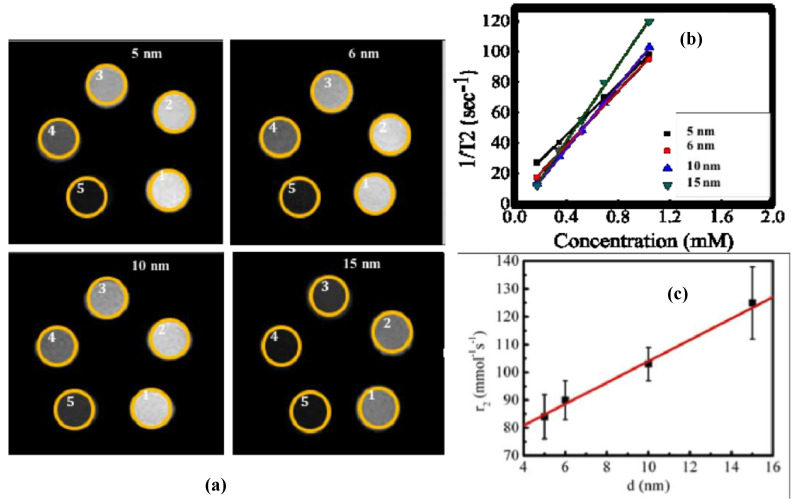
MRI data of CS-coated MnFe_2_O_4_ spinel ferrites nanoparticles. (**a**) Images acquired at TE of 14 ms (particle sizes of 5, 6, 10, and 15 nm) for CS-coated MnFe_2_O_4_ spinel ferrites nanoparticles with different values of concentrations (0.17, 0.34, 0.51, 0.68 and 1.03 mM) inside the five tubes in each image *1* to *5* represents lower to higher concentration of the nanoparticles in solution demonstrating contrast agents at different particle sizes. (**b**) Absolute R_2_ (or 1/T_2_) mapping images for particle sizes of 5, 6, 10, and 15 nm at different concentrations. (**c**) Nanoparticle size dependence of relaxivity (*r*_2_) of CS-coated MnFe_2_O_4_ nanoparticles.

**Figure 14 nanomaterials-10-02297-f014:**
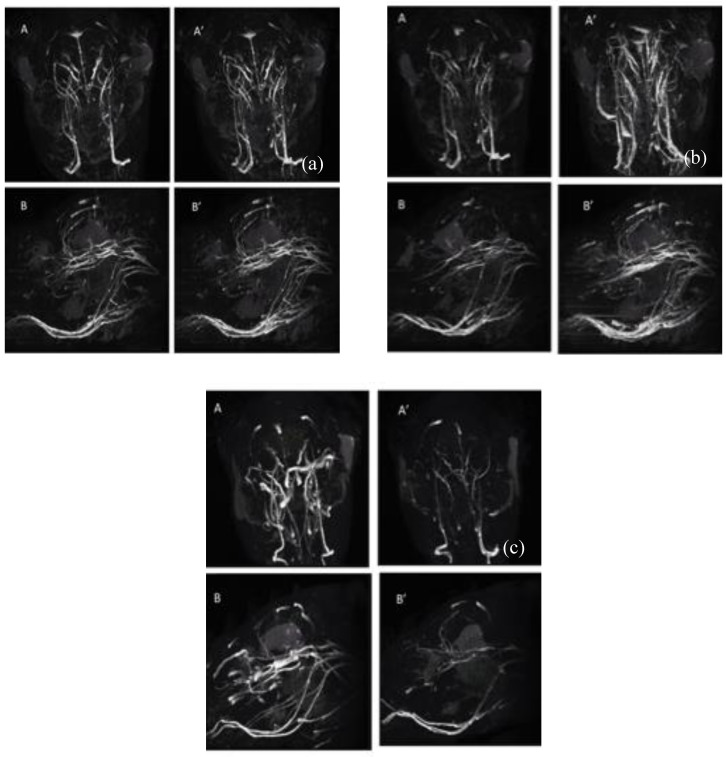
MRA with TOF 3D in vivo demsontrated in rat brain. Images from maximum intensity projection (MIP) are shown, where **A** and **A**’ represents a slice of MIP with and without contrast agents at 3° from the horizontal position, **B** and **B**’ represents with and without contrast agents at the sagittal position for the nanoparticle size of (**a**) 6 nm, (**b**) 10 nm, and (**c**) 15 nm.

**Table 1 nanomaterials-10-02297-t001:** Hyperfine parameters of MnFe_2_O_4_ nanoparticles with different pH without magnetic field and at room temperature.

Particle Size (nm)	FWHM	Isomer Shift mm/s	Quadruple Splitting mm/s	Hyperfine Field kG	Area
5	0.390	0.258	0.700	0	0.450
	0.373	0.428	0.700	0	0.450
	0.800	0.319	0.137	453	0.100
6	0.447	0.277	0.700	0	0.420
	0.413	0.412	0.700	0	0.430
	0.800	0.396	0.264	432	0.150
10	0.410	0.260	0.700	0	0.370
	0.380	0.420	0.700	0	0.380
	0.800	0.330	0.295	450	0.250
15	0.708	0.319	0.700	0	0.370
	0.426	0.370	0.700	0	0.380
	0.718	0.336	0.470	488	0.250

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
