# Peer review of "Manganese Ferrite Nanoparticles (MnFe2O4): Size Dependence for Hyperthermia and Negative/Positive Contrast Enhancement in MRI"

_nanomaterials, 2020, doi:10.3390/nano10112297_

Round 1
Reviewer 1 Report
The paper is aimed at investigating the efficacy of synthesized chitosan-coated MnFe2O4 nanoparticle as a negative/positive MRI contrast agent in the rat model. The relevance and importance of the topic are important. The audience of the journal will definitely find it interesting to read about a nontoxic, biocompatible, and biodegradable MnFe2O4 as MRA contrast agents, and also and can be a starting point for future studies for others researchers teams. The originality and novelty of the article are adequate. The high quality standards of the journal are adequately met, as the investigation methods are adequate, the results are relevant and the conclusions are pertinent. The methods are well chosen and represent adequate ways of achieving the main goal. I have only one objection to used cytotoxicity tests. I consider that there are other methods that are much more relevant and objective than the one used in this manuscript. Maybe this section should be rethought for the accuracy of the results.
Author Response
Reviewer 1
Comment R1.1 - The paper is aimed at investigating the efficacy of synthesized chitosan-coated MnFe2O4 nanoparticle as a negative/positive MRI contrast agent in the rat model. The relevance and importance of the topic are important. The audience of the journal will definitely find it interesting to read about a nontoxic, biocompatible, and biodegradable MnFe2O4 as MRA contrast agents, and also and can be a starting point for future studies for others researchers teams. The originality and novelty of the article are adequate. The high quality standards of the journal are adequately met, as the investigation methods are adequate, the results are relevant and the conclusions are pertinent. The methods are well chosen and represent adequate ways of achieving the main goal.
Response R1.1 - We thank R1 for overall positive comments on our work.
R1.2- I have only one objection to used cytotoxicity tests. I consider that there are other methods that are much more relevant and objective than the one used in this manuscript. Maybe this section should be rethought for the accuracy of the results.
Response R1.2 - We realize the culture studies for cytotoxicity test is limiting. However when the culture studies are taken in the context of the doses used in the in vivo MRI results shown, then we believe the cytotoxicity tests are relevant. However, the main PI’s laboratory in Bangladesh is creating resources for in vivo cytotoxicity tests. We have added a statement in the “Cytotoxicity Analysis” which states the following:
“These culture studies should be taken in the context of the doses used in the in vivo MRI Analysis, but future in vivo cytotoxicity tests would further establish the lethal doses for these nanomaterials”.
Reviewer 2 Report
The article entitled "Manganese ferrite nanoparticles (MnFe2O4): size dependence for hyperthermia and negative/positive contrast enhancement in MRI" is presented the results related to the synthesis of size-tuning MnFe2O4 nanoparticles coated with polysaccharide chitosan and the possibility of their practical application in biomedicine. The obtained results showed that the synthesized nanoparticles undergo successful surface modifications and are noncytotoxic and viable for cell lines. The authors clearly demonstrated that the nanoparticles can be used for hyperthermia and as a negative/positive MRI contrast agent and also revealed better size/concentrations of the nanoparticles for the corresponding applications. Thus, the article is interesting and relevant. My comments are listed below.
1) In line 9, the affiliation number is incorrect.
2) In lines 179 and 183, the symbol "d" denotes both interplanar spacing and particle size. It would be better to use various symbols (eg "D" for crystallite size).
3) The number of labels shown on y-axes in Figures 1 (d), 2 (A), 5 (b, d, e) seems excessive and makes it difficult to the visual perception of the figures.
4) In Figure 3, image 3d is marked as 3e. Also, the use of green color in Figure 3d is an unsuccessful choice because it's hard to see. In addition, in Figure 5 are shown only images (a-c), which does not coincide with the caption to the figure.
Author Response
Comment R2.1 - The article entitled "Manganese ferrite nanoparticles (MnFe2O4): size dependence for hyperthermia and negative/positive contrast enhancement in MRI" is presented the results related to the synthesis of size-tuning MnFe2O4 nanoparticles coated with polysaccharide chitosan and the possibility of their practical application in biomedicine. The obtained results showed that the synthesized nanoparticles undergo successful surface modifications and are noncytotoxic and viable for cell lines. The authors clearly demonstrated that the nanoparticles can be used for hyperthermia and as a negative/positive MRI contrast agent and also revealed better size/concentrations of the nanoparticles for the corresponding applications. Thus, the article is interesting and relevant. My comments are listed below.
Response R2.1 - We thank R2 for overall positive comments on our work.
Comment R2.2 - In line 9, the affiliation number is incorrect.
Response R2.2 - Corrected as indicated.
Comment R2.3 - In lines 179 and 183, the symbol "d" denotes both interplanar spacing and particle size. It would be better to use various symbols (eg "D" for crystallite size).
Response R2.3 – We used the notation dip for interplanar spacing and highlighted in the text.
Comment R2.4 - The number of labels shown on y-axes in Figures 1 (d), 2 (A), 5 (b, d, e) seems excessive and makes it difficult to the visual perception of the figures.
Response R2.4 - We have modified Figures 1, 2, and 5 as advised.
Comment R2.5 - In Figure 3, image 3d is marked as 3e. Also, the use of green color in Figure 3d is an unsuccessful choice because it's hard to see. In addition, in Figure 5 are shown only images (a-c), which does not coincide with the caption to the figure.
Response R2.5 - We have modified Figures 3 and 5 as advised.